# Solvent concentration at 50% protein unfolding may reform enzyme stability ranking and process window identification

Frieda A. Sorgenfrei [1], Jeremy J. Sloan [2], Florian Weissensteiner [1,3], Marco Zechner[1], Niklas A. Mehner [2], Thomas L. Ellinghaus [2], Doreen Schachtschabel[2], Stefan Seemayer[2] ✉ & Wolfgang Kroutil [1,3,4,5] ✉

As water miscible organic co-solvents are often required for enzyme reactions to improve e.g., the solubility of the substrate in the aqueous medium, an enzyme is required which displays high stability in the presence of this co-solvent. Consequently, it is of utmost importance to identify the most suitable enzyme or the appropriate reaction conditions. Until now, the melting temperature is used in general as a measure for stability of enzymes. The experiments here show, that the melting temperature does not correlate to the activity observed in the presence of the solvent. As an alternative parameter, the concentration of the co-solvent at the point of 50% protein unfolding at a specific temperature T in short $c_{U_{50}}^T$ is introduced. Analyzing a set of ene reductases, $c_{U_{50}}^T$ is shown to indicate the concentration of the co-solvent where also the activity of the enzyme drops fastest. Comparing possible rankings of enzymes according to melting temperature and $c_{U_{50}}^T$ reveals a clearly diverging outcome also depending on the specific solvent used. Additionally, plots of $c_{U_{50}}$ versus temperature enable a fast identification of possible reaction windows to deduce tolerated solvent concentrations and temperature.

Employing organic co-solvent in biocatalytic reactions gives the opportunity to tune the reaction properties and address a variety of challenges[1–3]. For instance, water miscible organic co-solvents are often applied, e.g., to improve the solubility and availability of substrates in an aqueous environment[2,4,5]. Thereby, it is essential that the biocatalyst tolerates the organic solvent while maintaining its activity. Consequently, evaluating the stability of biocatalysts is of utmost importance to identify the most suitable catalyst. For this purpose, a descriptive parameter for enzyme stability in organic solvents is needed to provide a guideline for the identification of the most suitable enzyme or variant e.g., during screening of enzyme libraries and enzyme engineering campaigns[6–9]. Analyzing the stability via activity measurements is very time consuming[10] and will become unfeasible with an increasing number of enzymes if no suitable high-throughput

assay is available[11]. As an alternative, it is common in literature to evaluate the stability by measuring the melting temperature[12]. It is defined as the temperature at which equal amounts of protein are folded and unfolded under the given conditions. Different methods to determine the melting temperature have been described which all share the principle that a sample is gradually heated. The state of the enzyme can be monitored for example (1) by measuring the fluorescence signal of internal tryptophane residues, (2) by using a fluorescent dye creating a change in fluorescence upon interaction with the unfolding protein or (3) e.g., in the case of ene reductases following the change of fluorescence of the cofactor FMN[13]. Many examples of enzymes with elevated thermal stability were identified either from extremophiles possessing an inherently elevated tolerance to temperature[14–16] or by enzyme engineering[7]. Additionally, a number of

[1]Austrian Centre of Industrial Biotechnology c/o University of Graz, Heinrichstrasse 28, 8010 Graz, Austria. [2]BASF SE, Carl-Bosch-Strasse 38, 67056 Ludwigshafen, Germany. [3]Department of Chemistry, University of Graz, NAWI Graz, Heinrichstrasse 28, 8010 Graz, Austria. [4]BioTechMed Graz, 8010 Graz, Austria. [5]Field of Excellence BioHealth, University of Graz, 8010 Graz, Austria. ✉e-mail: stefan.seemayer@basf.com; wolfgang.kroutil@uni-graz.at

computational tools have been published either to predict thermal stability[17,18] or to suggest and design variants with improved thermal properties[19–24]. For instance, the program FRESCO was successfully employed on different enzyme classes, covering hydrolases[19,23], Bayer-Villinger monooxygenases[20], alcohol dehydrogenases[21] and transaminases[22] to optimize their thermal properties. Besides this, other strategies including ancestral sequence reconstruction[24], sequence-, structure-[25,26] and molecular dynamics[27] guided design yielded enzymes with increased thermal resistance. There are also examples of improving the thermal tolerance of ene reductases e.g., by rational design[25,26]. In general, a raised melting point or thermo-tolerance was found to go hand in hand with an increased tolerance to organic solvents[22–27]. Recent studies aimed to understand the influence of co-solvents on enzyme activity on a structural basis[28–30] to be able to predict candidates for protein engineering[31,32]. An alternative para-meter for the selection of an organic solvent for a biotransformation is the denaturation capacity of a solvent as guidance across different enzymes[33].

Although the melting temperature may provide a guideline to improve stability toward an organic solvent, it cannot serve as a quantitative measure for enzyme activity. The melting temperature cannot be correlated to a co-solvent concentration up to which the enzyme is active. Consequently, alternative strategies are required to simplify assessing the organic solvent tolerance of enzymes. In selected cases it has already been reported that enzymes showed actually higher activity at low co-solvent concentration compared to reaction conditions in the absence of co-solvent[30,34]. The reasons for this may be manifold, including structural changes or increased mobility in the enzyme through modified hydrogen bond networks[34].

Here, we evaluate enzyme activity in the presence of organic solvents in comparison to the melting temperature. Subsequently, we suggest an alternative parameter ($c_{U_{50}}^{T}$) to describe the stability of an enzyme in the presence of a defined organic co-solvent and show that this parameter is linked to activity. A ranking of enzymes/variants was found to differ significantly depending on the measure used (melting temperature or $c_{U_{50}}^{T}$ for each solvent).

## Results and discussion

For this study, ene-reductases (EREDs) using the cofactors FMN and NAD(P)H as cosubstrate were selected as model catalysts possessing high relevance for biocatalysis[35–41]. EREDs are well known to reduce activated C=C double bonds[42–44] and more recently were used for asymmetric proton transfer[45] or oxime reduction[46].

A small library of 13 EREDs[41,47] was selected covering a range of wild type EREDs (a table with all names and organisms of origin can be found in Supplementary Table 1). The EREDs were selected considering the phylogenetic tree of EREDs (Fig. 1a) ensuring that most branches were covered. A pairwise sequence distance analysis of the selection showed that they cluster into two main similarity groups beside the three EREDs ChrOYE1, LacER and YqiG (Fig. 1b). The percentage pairwise sequence identities within this selection ranges from around 20 to 50 with the extreme example of XenB and PpXenB with a percentage pairwise sequence identity of 88. These 13 representative EREDs were hetero-logously expressed and used as purified enzymes (Supplementary Fig. 1).

### Thermal stability of EREDs in the presence of increasing con-centration of co-solvent

As a common thermal stability measure, the melting temperature of the selected EREDs was recorded in sodium phosphate buffer (50 mM, pH 7.4). The melting temperatures in the absence of co-solvent were found to be between 40.7 ± 0.3 °C for NerA and above 90 °C for TsOYE (Supplementary Fig. 2 and Supplementary Table 2). The values mea-sured were in line with previous studies e.g., for XenA and YqjM (49.0 ± 0.0 °C and 50.9 ± 0.3 °C, respectively)[48,49]. As a next step, the melting temperatures of each enzyme at varied concentration [5 to 30% (v/v)] of five organic co-solvents (DMSO, methanol, ethanol, 2-propanol and n-propanol) in buffer was measured. Comparing the change of melting temperature with increasing solvent concentration showed a similar picture for all tested EREDs (Fig. 2a and Supple-mentary Fig. 3). On the one hand, in all cases DMSO led to the smallest change of melting temperature at increasing solvent concentration. On the other hand, the primary alcohol n-propanol as co-solvent decreased the melting temperature the most among the tested co-solvents. It is worth to emphasize that the impact of the solvents on the

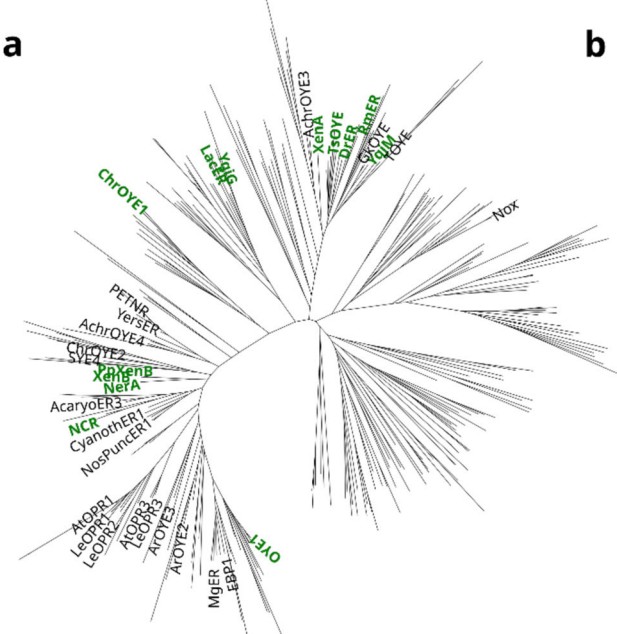

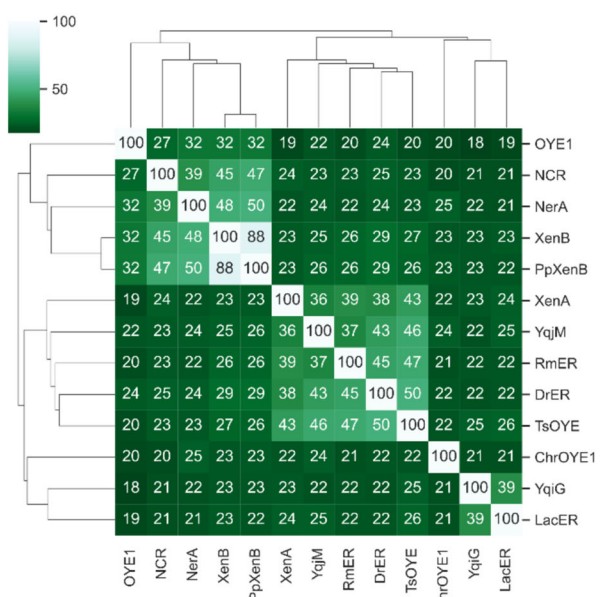

**Fig. 1 | Phylogenetic tree and pairwise sequence identities of ene reductases.**
**a** In the phylogenetic tree the 13 representative EREDs selected for this study are labeled in green while other known EREDs are labeled in black. **b** The percentage pairwise sequence identities of the selected EREDs cluster into two main similarity groups. The color scale ranges from 0% (dark green) to 100% (white) identity.

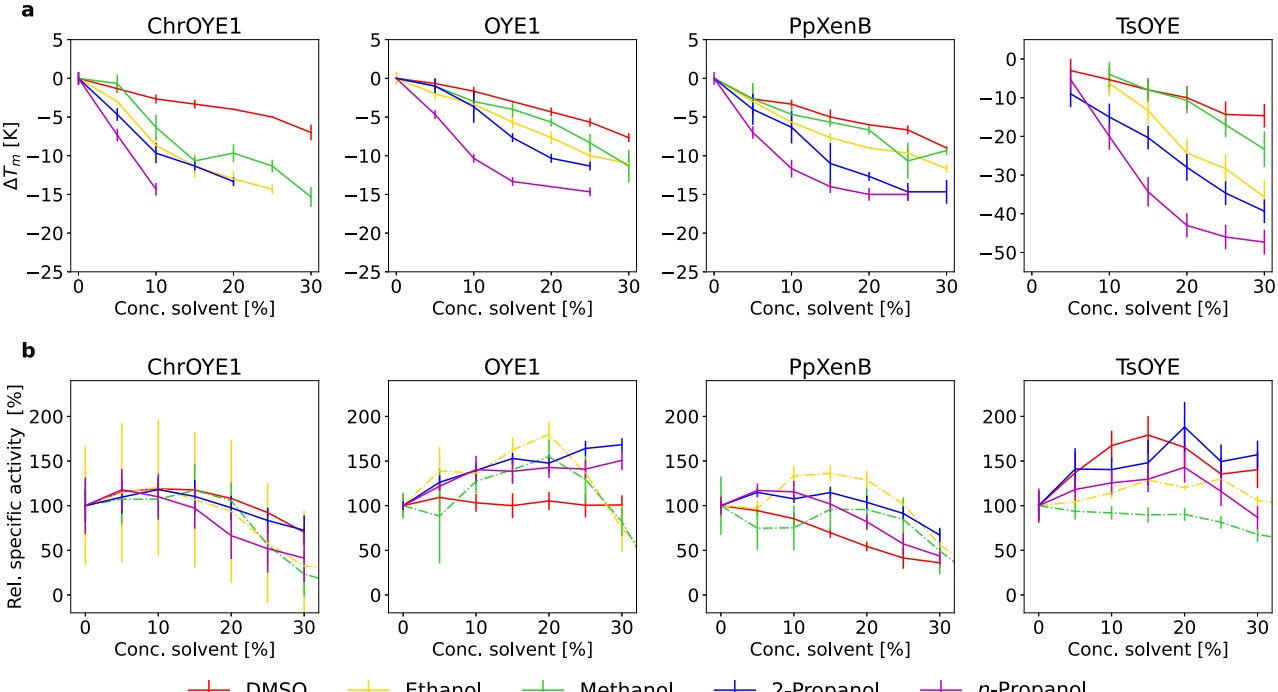

**Fig. 2 | Co-solvent concentration-dependent changes in ERED stability and activity. a** Change of melting temperature $\Delta T_m$ and **b** relative (Rel.) specific activity of four selected EREDs at increasing co-solvent concentration (% v/v) (Conc.) [dimethyl sulfoxide (DMSO) (red), ethanol (yellow), methanol (green), 2-propanol (blue), and *n*-propanol (magenta)]. The complete set of results can be found in Supplementary Figs. 3 and 4. The melting temperature was measured in 50 mM sodium phosphate buffer pH 7.4 and the mean change of melting temperature induced by the co-solvents of three replicates is given. The error bar marks the standard error of the mean using Gaussian error propagation. The specific initial activity was recorded under the same conditions using cyclohex-2-enone (10 mM) as substrate following the decrease of NAD(P)H (0.2 mM) at 340 nm and room temperature (i.e., 20–24 °C for DMSO, 2-propanol and *n*-propanol) and 25 °C for ethanol and methanol. Given is the mean of three replicates. The values are normalized by setting the activity measured without co-solvent (normal conditions) to 100%. Values above 100 indicate an increased activity while values below 100 indicate a decreased activity relative to the activity under normal conditions. The error bar marks the standard error of the mean using Gaussian error propagation. The relative specific activities calculated from measurements at 25 °C given here are shown again as part of Figs. 4b and 5b. Measured values are connected by lines to facilitate reading of the figure and do not represent measured values. All raw data that was used to obtain the derived values $\Delta T_m$ and relative specific activity can be found in Supplementary Tables 5 and 6.

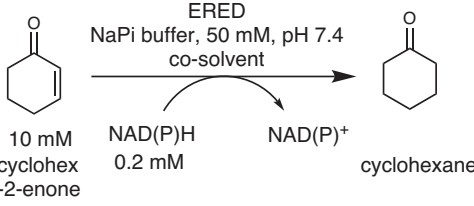

**Fig. 3 | Model reaction for the determination of ERED specific initial activity by following NAD(P)H depletion at 340 nm.** EREDs (5–30 μg ml⁻¹) were analyzed using model substrate cyclohex-2-enone (10 mM) using the following reaction conditions: NAD(P)H (0.2 mM) in sodium phosphate buffer (50 mM, pH 7.4), a reaction volume of 200 μl, in the absence or presence of varied concentration of co-solvents [DMSO, methanol, ethanol, 2-propanol, *and n*-propanol at 0–45% (v/v)].

melting temperature was always in the same order for all tested enzymes: DMSO led to the smallest decrease, followed mostly by methanol, ethanol, 2-propanol and finally *n*-propanol caused the highest decrease of melting temperature. This seems to indicate that the thermal destabilization depends on the nature of the co-solvent rather than on the enzyme. Furthermore, it was observed that while NerA has the lowest $T_m$ it also showed the smallest $\Delta T_m$, while TsOYE is the other end of the spectrum displaying the largest $\Delta T_m$ (see TsOYE, $T_{m,nat} > 90$ °C). Thus, the absolute decrease of melting temperature observed of an enzyme with high native melting temperature was larger. This observation was particularly pronounced in the presence of 10% (v/v) *n*-propanol.

## Specific activity at varied concentration of organic solvent

In a next step, the specific activity was measured for the same co-solvent concentrations as for the thermal unfolding. For this purpose, the ERED catalyzed reduction of cyclohex-2-enone to cyclohexanone was followed by measuring the depletion of NAD(P)H spectrophotometrically (Fig. 3)[32,47,50,51].

Comparing the influence of varying concentrations of solvent on the activity (Fig. 2b and Supplementary Fig. 4) with $\Delta T_m$ (Fig. 2a and Supplementary Fig. 3) reveals a clear difference between the enzymes. While shape and order of the melting temperature curves in the presence of increasing concentrations of co-solvent followed the same succession, this was not the case for the activity (Fig. 2). The melting temperature decreased under all tested conditions, whereas the activity was in some cases boosted, or not influenced and in other cases decreased, thus the solvents' effects on the enzymes differed clearly. Although, the addition of small amounts of co-solvent led to an increased specific activity in selected cases; high concentrations of co-solvent mostly led to a loss of activity. A correlation analysis of the relative specific activity with the change of melting temperature demonstrates that there is no general correlation between the observed thermal destabilization and the measured activity (Pearson correlation coefficient <0.15) (Supplementary Fig. 5). Even though, there are many examples[22–24,26,27] showing that an enzyme engineered for thermal stability also showed higher co-solvent tolerance, this data suggests that melting temperature or the change of melting temperature by co-solvent is not a good measure to draw conclusions on the activity of an enzyme in a specific solvent.

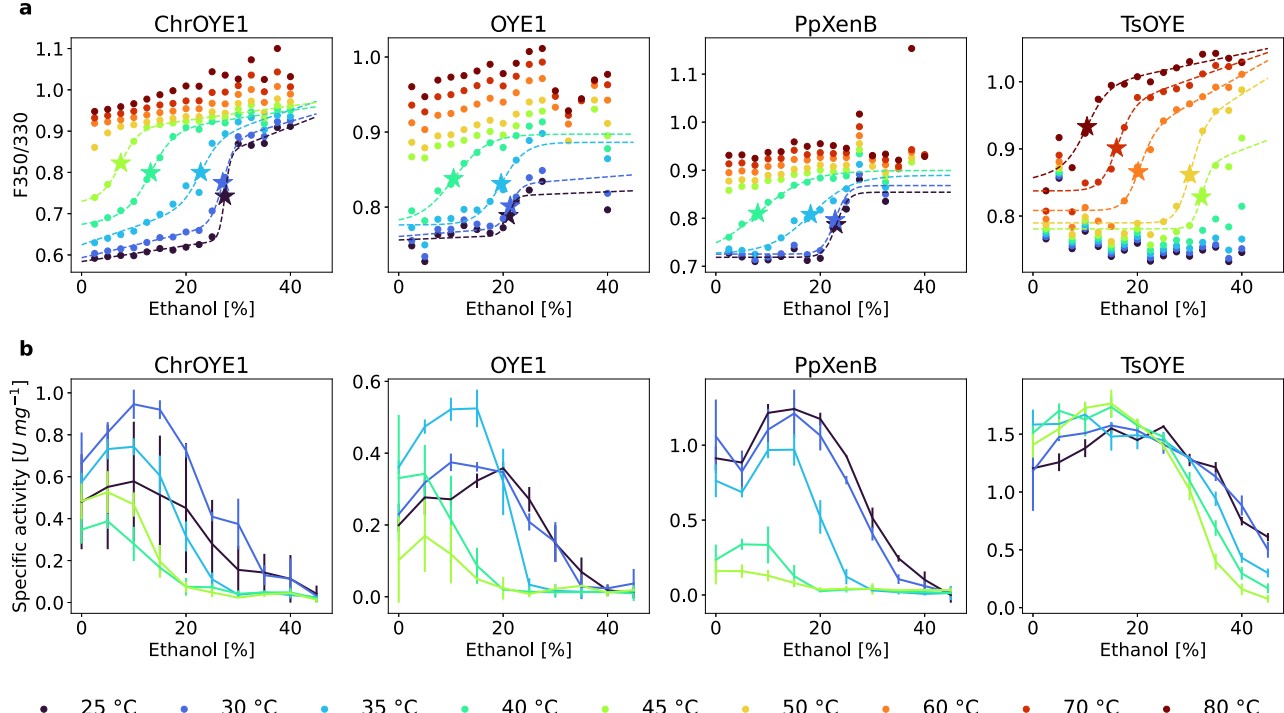

**Fig. 4 | The influence of ethanol on unfolding and comparison to specific activity. a** The unfolding of EREDs induced by the presence of increasing amounts of ethanol at different temperatures [color range from dark blue (25 °C) to dark red (80 °C)] measured by nanoDSF [given as the ratio of the fluorescence at 350 and 330 nm (F350/330)]. **b** The initial activity recorded at the same co-solvent and temperature conditions. A drop of activity can be found close to the concentration of unfolding. The dashed lines in (**a**) represent fits of the two-state model of unfolding Eq. (1) to the recorded data. The star marks the co-solvent concentration of unfolding $c_{U_{50}}^T$ at the specific temperature. The solid lines in (**b**) connect the measured values of each reaction temperature for visual guidance. The mean activity was calculated from three replicates and the error bar marks the standard error of the mean using Gaussian error propagation.

Frequently, increased solubility of the substrate in the presence of a co-solvent and with that the better availability of the substrate to the enzyme is mentioned as a reason why the addition of an organic co-solvent influences the activity positively[52–54]. However, for our model reaction a better solubility of the substrate does not explain the boosted activity, because the solubility of cyclohexanone in water (234 mM at 20 °C) is not limiting under the conditions used[55]. In contrast, the decreased melting temperature of all tested enzymes showed that they were thermally destabilized by the presence of the co-solvent, nevertheless some EREDs showed actually higher initial rates under the respective condition. Probably as previously suggested[34], an increased flexibility and higher dynamics induced by the presence of the co-solvent may for example induce an allosteric shift of the conformational space[56,57] that is beneficial for the catalysis.

## Co-solvent induced unfolding as alternative stability measure

As the melting temperatures and the activity observed in the presence of various co-solvents did not correlate, an alternative stability measure to the commonly used melting temperature is required. We suggest, that a potential parameter to judge the solvent tolerance of an enzyme may be deduced from the unfolding induced by the co-solvent. Similar to the melting temperature, the concentration of the co-solvent can be determined at which half of the protein is folded and half is unfolded, thus the free energy of folding is zero $\Delta G_{folding}^{solv, T=const.}(c_{solv}) = 0$. This concentration is abbreviated here $c_{U_{50}}^T$ (concentration of 50% unfolded), where T stands for the temperature. To obtain $c_{U_{50}}^T$ values for enzyme/co-solvent combinations, samples containing the same enzyme at varied concentrations of solvent were followed via nano differential scanning fluorimetry (nanoDSF) experiments, thus measuring the unfolding at increasing temperature. NanoDSF is used to track the fluorescent signal arising from the internal tryptophan/tyrosine residues and has previously been used to identify melting points

of enzyme libraries[58]. From multiple nanoDSF experiments the $c_{U_{50}}^T$ can be extracted by plotting the unfolding at a specific temperature versus solvent concentration (Figs. 4a and 5a). Consequently, the co-solvent induced unfolding curves were measured for a range of temperatures for ethanol and methanol as the solvents. All four enzymes displayed curves that start in the folded state and upon the addition of co-solvent a transition to the unfolded state can be observed.

These data also indicate that the co-solvent induced unfolding is also dependent on temperature. By increasing the temperature of the sample, the concentration of co-solvent needed for unfolding is shifted to lower co-solvent concentrations. Unfolding was recorded for all 13 EREDs, the complete data set can be found in Supplementary Figs. 6 and 7. For some enzymes this method did not give an analyzable unfolding curve (e.g., LacER, see Supplementary Figs. 6 and 7), which was attributed for example to tryptophan residues with little change in their local environment during the unfolding process. In order to quantitatively analyze the successfully recorded unfolding data, $c_{U_{50}}^T$ was determined using the literature known empirical extended two-state unfolding model described by Eq. (1)[59,60].

In this model of unfolding [Eq. (1)], $c_{solv}$ is the concentration of the co-solvent and $c_{U_{50}}^T$ represents the co-solvent concentration at a specific temperature $T$ where half of the protein is unfolded. $m_{folding}$ is a proportionality constant of the amount of co-solvent and the free energy of folding $\Delta G_{folding}$, while $R$ and $T$ are the universal gas constant and the temperature, respectively. $\alpha$ and $\beta$ are correction factors for the linear trends of the unfolded [U] and folded [F] signal which are often observed in spectroscopic data[59,60]:

$$F_{350nm/330nm} = \frac{\alpha_F + \beta_F c_{solv} + (\alpha_U + \beta_U c_{solv}) \exp\left(\frac{m_{folding}}{RT}\left(c_{solv} - c_{U_{50}}^T\right)\right)}{1 + \exp\left(\frac{m_{folding}}{RT}\left(c_{solv} - c_{U_{50}}^T\right)\right)} \quad (1)$$

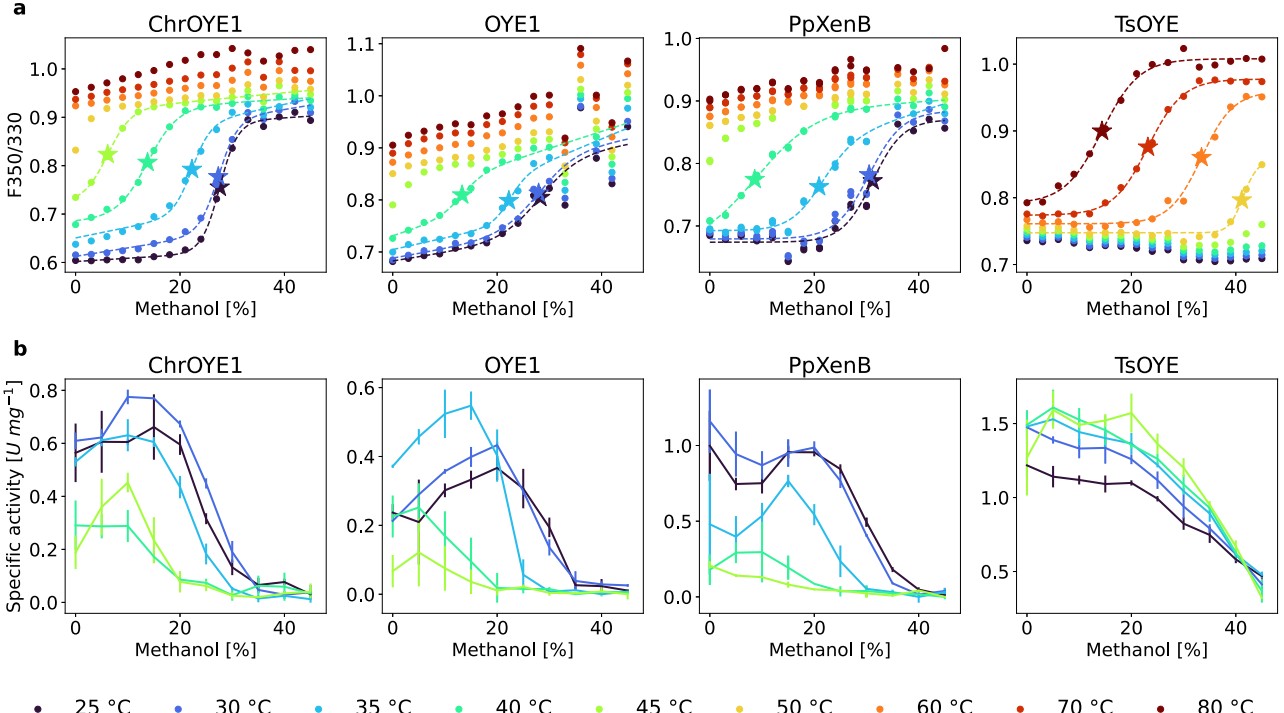

**Fig. 5 | The influence of methanol on unfolding and the comparison to activity.**
**a** The unfolding of EREDs induced by the presence of increasing amounts of methanol at different temperatures [color range from dark blue (25 °C) to dark red (80 °C)] measured by nanoDSF [given as the ratio of the fluorescence at 350 and 330 nm (F350/330)]. **b** The initial activity recorded at the same co-solvent and temperature conditions. A drop of activity can be found close to the concentration of unfolding. The dashed lines in (**a**) show fits of the two-state model of unfolding Eq. (1) to the recorded data. The star marks the co-solvent concentration of unfolding $c_{U_{50}}^T$ at the specific temperature. The solid lines in (**b**) connect the measured values of each reaction temperature for visual guidance. The mean activity was calculated from three replicates and the error bar marks the standard error of the mean using Gaussian error propagation.

Fitting this model to the above-described unfolding data gives $c_{U_{50}}^T$, which is marked in the figures with a star (Figs. 4a and 5a and Supplementary Figs. 6 and 7). We hypothesize that this $c_{U_{50}}^T$ value may indicate the co-solvent concentration where the enzyme loses its activity, and thus be of high relevance to describe enzyme stability in a specific organic solvent.

For this purpose, the activities were measured at the same co-solvent conditions and the same temperatures in the range of 25–45 °C (Figs. 4b and 5b). Intriguingly, a loss of activity at a co-solvent concentration around $c_{U_{50}}^T$ was observed showing that there is indeed very likely a link between $c_{U_{50}}^T$ and the concentration at which the activity drops fastest (Figs. 4 and 5). This observation can be rationalized since the correct structural fold of an enzyme is essential for its function. More examples can be found in the Supplementary Information by comparing Supplementary Figs. 6–9. Consequently, a model was developed to link unfolding and activity and to describe the observed co-solvent dependence of the activity.

**Model for the dependence of activity on solvent concentration**
It is fair to assume that the correct structural fold of an enzyme is essential for its function. Therefore, we hypothesized that the basic two-state model [Eq. (2)] of unfolding which is derived from thermodynamics and kinetics and describes the ratio of unfolded to total protein (folded plus unfolded), is a good starting point to develop an empirical-mathematical model to describe the activity in dependence of the concentration of the co-solvent:

$$\frac{[U]}{[U]+[F]} = \frac{\exp\left(\frac{m_{folding}}{RT}\left(c_{solv} - c_{U_{50}}^T\right)\right)}{1 + \exp\left(\frac{m_{folding}}{RT}\left(c_{solv} - c_{U_{50}}^T\right)\right)} \quad (2)$$

While Eq. (2) can describe two states, it cannot describe the boosted activity in the presence of low amounts of co-solvent observed for various enzymes in this study and in literature[61]. Similar to the correction that is commonly applied to the two-state model of unfolding [see Eq. (1)] we introduced a Gaussian term to adapt it for the description of *Activity* as shown in Eq. (3):

$$Activity = \xi \frac{1}{\sigma\sqrt{2\pi}} \exp\left(-0.5\left(\frac{c_{solv} - c_{A_{max}}}{\sigma}\right)^2\right)$$
$$+ \nu \frac{\exp\left(-\frac{m_{folding}}{RT}\left(c_{solv} - c_{A_{50}}\right)\right)}{1 + \exp\left(-\frac{m_{folding}}{RT}\left(c_{solv} - c_{A_{50}}\right)\right)} \quad (3)$$

This combined term can describe the boosted activity observed at lower co-solvent concentrations before the onset of unfolding leads to a drop of activity. $c_{A_{50}}$ can be interpreted as the concentration of co-solvent at which the loss of activity is largest. $c_{A_{max}}$ gives the ideal concentration of co-solvent at which the activity is maximal and $\sigma$ gives evidence on how narrow the tolerated concentration range is. $\xi$ and $\nu$ are correction terms necessary for the scaling of the Gaussian and the two-state term. As the concentration of co-solvent increases, the boosting effect reaches a maximum and then starts to decrease until the unfolding of the enzyme dictates the loss of activity of the system. In the cases where no significant increase of activity is observed the influence of the Gaussian term becomes very small and the main influence during data fitting comes from the unfolding term.

Next, Eq. (3) was fitted to the activity data reported in Figs. 4b and 5b. The fitted curves showed that the model described the recorded activity across all tested EREDs and reaction temperatures in the presence of ethanol and methanol in a concentration ranging from 0 to 45% (v/v) very well (Fig. 6 and Supplementary Figs. 10 and 11).

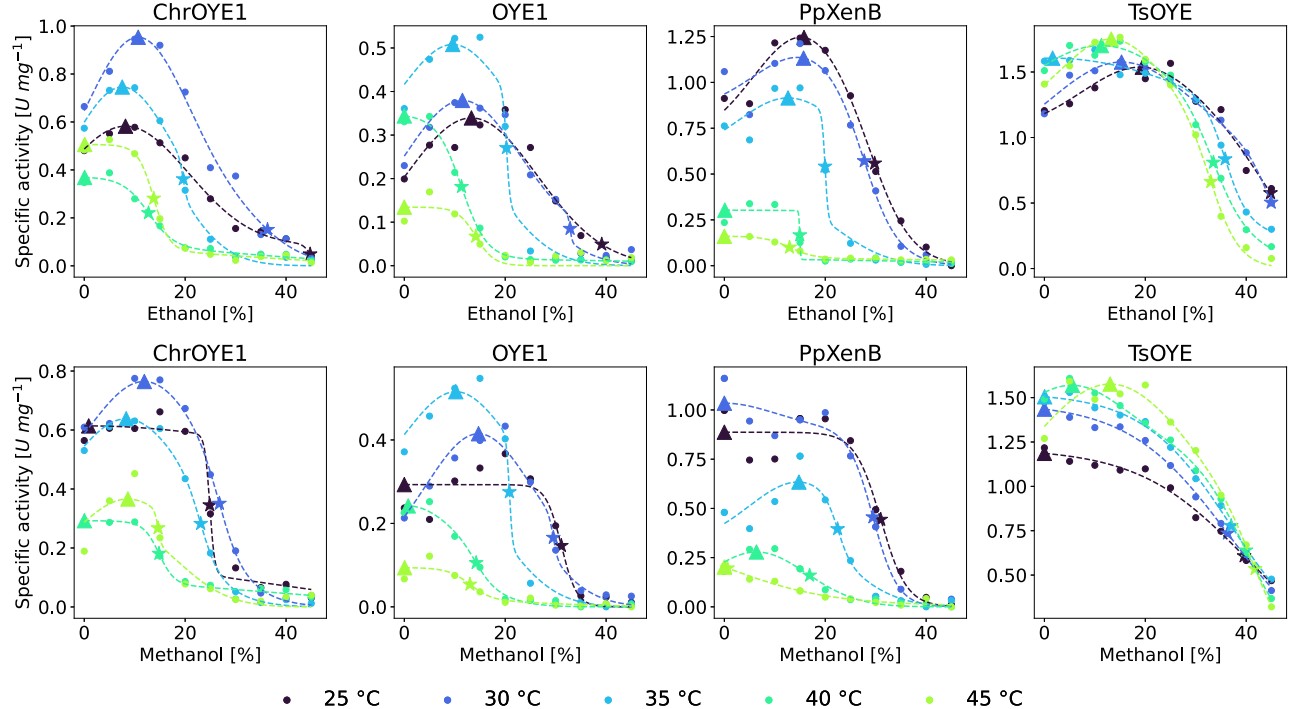

**Fig. 6 | Activity-solvent model [Eq. (3)] fitted to specific activity data of EREDs.** The initial specific activity data of four EREDs in the presence of increasing amount of ethanol and methanol fitted with the activity-solvent model Eq. (3). The fits are displayed as dashed lines, the measured initial activity is displayed as circles and are also shown in Figs. 4 and 5. The color of the curve represents the reaction temperature. The stars mark $c_{A_{50}}$, the concentration of largest loss in activity and the triangles mark $c_{A_{max}}$ the concentration of maximal activity.

## Comparison of activity and unfolding related parameters $c_{A_{50}}$ and $c_{U_{50}}^T$

Comparing the fitted values for $c_{A_{50}}$ and $c_{U_{50}}^T$ for a subset of enzymes and temperatures (Table 1) indicates that $c_{A_{50}}$ and $c_{U_{50}}^T$ are in general close together and that $c_{U_{50}}^T$ may consequently be used to estimate the concentration where enzymes lose their activity.

The correlation of the data of Table 1 was visualized together with the data at up to five temperatures between 25 and 45 °C (Supplementary Fig. 12). A correlation analysis of all determined $c_{A_{50}}$ and $c_{U_{50}}^T$ values demonstrated the dependence of these two parameters

(Pearson correlation coefficient ethanol: 0.724 and methanol: 0.672). For the analysis, all enzymes and conditions were considered, for which both the two-state unfolding model and the activity model were successfully fitted to the respective experimental data. In the case of YqiG no activity profile could be recorded above 40 °C due to low activity (Supplementary Figs. 10 and 11). For TsOYE the unfolding occurred mostly above 45 °C, while the activity assay was limited to reaction temperatures up to 45 °C. The $m_{folding}$ values of both fits do not show a correlation. In general, a correlation between the half unfolding concentration of co-solvent $c_{U_{50}}^T$ and the observed drop in activity could be demonstrated. Nevertheless, the correlation values may indicate how difficult it is to measure activities with low errors and that maybe a further improved activity model may be required.

## Impact of $c_{U_{50}}^T$ for selecting enzymes or variants and guideline for choosing suitable co-solvent and reaction temperature

As the concentration of co-solvent at steepest activity loss correlates with the concentration of 50% unfolding $c_{U_{50}}^T$ in a given solvent and at a given temperature, $c_{U_{50}}^T$ may represent a very useful parameter (1) for comparing the stability of enzymes or variants in solvents and consequently to identify the best candidate or (2) for identifying possible reaction conditions with respect to solvent concentration and temperature for a given enzyme.

When comparing stability of enzymes in the presence of water-miscible solvents using $c_{U_{50}}^T$ instead of the until now commonly used melting temperature $T_m$, we noticed that different candidates are at the top of the list (Table 2). This analysis refers to enzymes for which the unfolding at 30 °C and the melting temperature in the presence of 10% ethanol and methanol were determined. According to the $T_m$ in the absence of solvent the enzyme DrER would be the most stable, which would also be the case when measuring $T_m$ in the presence of 10% (v/v) ethanol or methanol. However, DrER is only the third most stable when $c_{U_{50}}^{30\,°C}$ was considered in the presence of ethanol.

**Table 1 | Fitted parameters for an example subset of unfolding and activity data**

| Enzyme | Solvent | $T_{reaction}$ [°C] | $c_{A_{max}}$ [% (v/v)] | $\sigma$ [% (v/v)] | $c_{A_{50}}$ [% (v/v)] | $c_{U_{50}}^T$ [% (v/v)] |
|---|---|---|---|---|---|---|
| ChrOYE1 | EtOH | 35 | 7.5 | 10.0 | 19.6 | 22.8 |
| DrER | EtOH | 35 | 0.0 | 12.1 | 14.9 | 17.4 |
| OYE1 | EtOH | 35 | 9.6 | 10.0 | 20.3 | 19.6 |
| PpXenB | EtOH | 35 | 12.6 | 10.0 | 20.0 | 18.1 |
| TsOYE | EtOH | 45 | 13.3 | 10.0 | 33.0 | 32.2 |
| YqiG | EtOH | 35 | 0.0 | 42.3 | 21.2 | 21.4 |
| ChrOYE1 | MeOH | 35 | 8.3 | 10.9 | 23.1 | 22.3 |
| DrER | MeOH | 35 | 0.0 | 57.4 | 18.5 | 18.4 |
| NCR | MeOH | 35 | 0.3 | 10.0 | 24.3 | 33.5 |
| OYE1 | MeOH | 35 | 10.2 | 10.0 | 20.8 | 22.2 |
| PpXenB | MeOH | 35 | 14.8 | 10.0 | 22.4 | 20.9 |
| YqiG | MeOH | 35 | 0.0 | 14.9 | 19.3 | 28.7 |

The parameters $c_{A_{50}}$, $c_{A_{max}}$ and $\sigma$ are from the fitting of the activity data to Eq. (3), $c_{U_{50}}^T$ is derived from Eq. (1).

$T_{reaction}$ experiment or reaction temperature, $c_{A_{50}}$ concentration of rapid loss of activity, $c_{A_{max}}$ concentration of maximal activity, $\sigma$ tolerance of deviation from best co-solvent concentration, $c_{U_{50}}^T$ concentration of half unfolding.

Measuring the $c_{U_{50}}^{30\,°C}$ in methanol, DrER was the least stable of the five listed enzymes. Actually, YqiG is the best candidate in the presence of ethanol and methanol. Even more pronounced is the difference for the enzyme PpXenB. According to the $T_m$, it is the least stable together with OYE1, however, according to the $c_{U_{50}}^{30\,°C}$ it is the second best in methanol. This analysis demonstrates that the order of best candidates completely changed compared to the parameter commonly used for ranking, the $T_m$. Furthermore, it is worth to note, that the order differs depending on the type of co-solvent investigated. This may not be surprising but until now this parameter was generally not analyzed during stability assessment. Due to these differences, $c_{U_{50}}^{T}$ is worth to be considered in the ranking of enzymes. Compared to the denaturation capacity parameter[33], our approach also allows for a ranking of different enzymes in dependence of the solvent and not a ranking of co-solvents only. Especially when looking for the best enzyme of a library or analyzing variants in a protein engineering campaign, it is of utmost importance to identify the best candidates efficiently.

Another application of $c_{U_{50}}^{T}$ may be to find the operational window of best suitable reaction conditions concerning temperature and solvent concentration for a given enzyme. Instead of labor-intensive activity tests in the two dimensions of temperature and co-solvent concentration, this information may be quickly deduced from the $c_{U_{50}}^{T}$. By analyzing $c_{U_{50}}^{T}$ versus reaction temperature (Fig. 7), suitable co-solvent concentration and reaction temperature combinations can

be identified by selecting the area below the $c_{U_{50}}^{T}$ versus $T_{reaction}$ curve. This strategy can reduce the screening efforts drastically because it enables to draw conclusions solely based on the analysis of unfolding data.

Finally, to show the applicability also for other solvents than the ones just mentioned (MeOH/EtOH), $c_{U_{50}}^{T}$ values were also measured for selected EREDs in DMSO, DMF and $n$-propanol (Table 3, Supplementary Table 4 and Supplementary Fig. 13). THF was also tested, but it was noticed, that it was not tolerated by the enzymes investigated. From the data obtained (e.g., entries 1–4) it can nicely be seen that the $c_{U_{50}}^{T}$ values change with the temperature, thus, the temperature has a clear impact. In general, the higher the temperature, the lower the $c_{U_{50}}^{T}$ value. Additional experiments also indicated the applicability to other enzymes like transaminases (entries 17–24). The two transaminases investigated, one (*S*)-selective one originating from *Arthrobacter citreus* (ArS) and the other (*R*)-selective one from an *Arthrobacter* sp. (ArR) possess different structural folds. Also for these enzymes $c_{U_{50}}^{T}$ values were successfully determined.

This study shows that the melting temperature is not an ideal measure to analyze enzyme stability in miscible water-co-solvent-mixtures. Furthermore, the melting temperature cannot be linked to a

### Table 2 | Ranking of enzyme stability for EREDs depending on the mode of analysis (melting temperature and unfolding concentration)

| ERED | Stability measure | | | | |
|------|-------------------|---|---|---|---|
| | $T_m$ | $T_m^{10\%}$ | $c_{U_{50}}^{30\,°C}$ | $T_m^{10\%}$ | $c_{U_{50}}^{30\,°C}$ |
| | w/o solvent | Ethanol | | Methanol | |
| DrER | + + + + + | + + + + + | + + + | + + + + + | + |
| YqiG | + + + + | + + + + | + + + + + | + + + + | + + + + + |
| ChrOYE1 | + + + | + | + + + + | + | + + |
| OYE1 | + + | + + + | + | + + + | + + + |
| PpXenB | + + | + + | + + | + + | + + + + |

The ranking shown here is based on the experimental values listed in Supplementary Table 3. The number of + indicate the ranking: The first rank, thus the most stable enzyme according to the type of measurement has five + (+ + + + +), the second four (+ + + +), the third three (+ + +), 4th (+ +) and 5th (+).

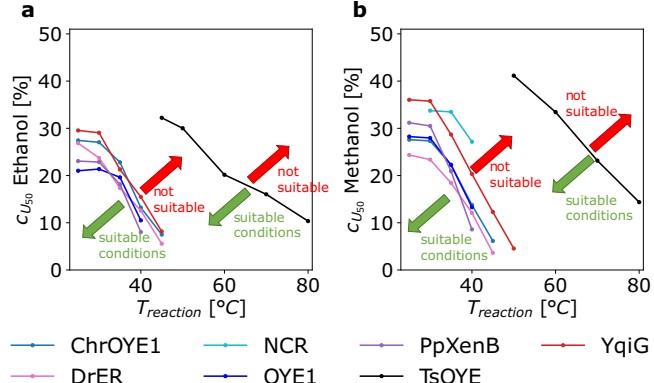

Fig. 7 | **Identification of suitable reaction conditions for reaction temperature and solvent concentration from a plot of $c_{U_{50}}^{T}$ at the respective temperature for various EREDs.** Suitable reaction conditions for **a** ethanol and **b** methanol are those at co-solvent and temperature values below the curve for each enzyme. The solid lines connect the measured values of each reaction temperature $T_{reaction}$ for visual guidance. $c_{U_{50}}^{T}$ is the concentration of half unfolding at a given temperature.

### Table 3 | Examples for $c_{U_{50}}^{T}$ values for various EREDs in further solvents as well as $c_{U_{50}}^{T}$ values for two transaminases (ArR, ArS)

| Entry | Enzyme | solvent | $T$ [°C] | $c_{U_{50}}^{T}$ [a] [% (v/v)] | Entry | Enzyme | solvent | $T$ [°C] | $c_{U_{50}}^{T}$ [a] [% (v/v)] |
|-------|--------|---------|----------|------------|-------|--------|---------|----------|------------|
| 1 | ChrOYE1 | DMF | 25 | 23 | 13 | TsOYE | DMSO | 70 | 41 |
| 2 | ChrOYE1 | DMF | 30 | 15 | 14 | TsOYE | nprop | 45 | 18 |
| 3 | ChrOYE1 | DMF | 35 | 13 | 15 | YqiG | DMF | 45 | 35 |
| 4 | ChrOYE1 | DMF | 40 | 11 | 16 | YqiG | nprop | 25 | 18 |
| 5 | ChrOYE1 | DMSO | 45 | 41 | 17 | ArR | nprop | 30 | 18 |
| 6 | ChrOYE1 | nprop | 30 | 10 | 18 | ArR | nprop | 35 | 18 |
| 7 | ChrOYE1 | nprop | 35 | 8 | 19 | ArR | nprop | 70 | 11 |
| 8 | ChrOYE1 | nprop | 40 | 7 | 20 | ArS | DMF | 60 | 31 |
| 9 | DrER | DMF | 30 | 23 | 21 | ArS | DMSO | 60 | 45 |
| 10 | DrER | DMSO | 30 | 33 | 22 | ArS | EtOH | 50 | 25 |
| 11 | DrER | nprop | 30 | 14 | 23 | ArS | MeOH | 45 | 47 |
| 12 | TsOYE | DMF | 60 | 37 | 24 | ArS | nprop | 45 | 23 |

The complete table can be found in the SI (Supplementary Table 4).
[a]The error of the $c_{U_{50}}^{T}$ was estimated to be about 2%.

solvent concentration threshold up to which an enzyme can be expected to be active and thus provides only limited information. As an alternative, $c_{U_{50}}^T$ – the concentration of a specific co-solvent [here in % (v/v)] at which half of the enzyme is folded and half is unfolded – is shown here to be correlated with activity and might therefore be better suitable to describe the behavior of an enzyme in buffer in the presence of a specific co-solvent. Exemplified for EREDs, it was demonstrated that when using melting temperature as selection criteria, not the best enzyme might be chosen for a specific co-solvent reaction condition. As $c_{U_{50}}^T$ is specific for a solvent-enzyme combination, the best enzyme or variant may differ depending on the investigated solvent. This is especially important for identifying, e.g., the best enzyme in a screening or the best variant in a protein engineering campaign. Furthermore, for a given enzyme, measuring $c_{U_{50}}^T$ at varied temperature allows to identify the operational window of the solvent concentration and the temperature. As the principles of this strategy are applicable for all enzymes having tryptophane/tyrosine in their sequence, this approach is not limited to EREDs and can be used for other types of enzymes, as we have shown here also for transaminases, and could have a major impact on the description of enzyme stability, selection of suitable enzymes from (commercial) libraries and in enzyme engineering campaigns, as well as on identifying operational windows for reactions. It is fair to assume that the applicability of the method can also be extended to any other organic compound and or inorganic component present, thus, allowing to determine the critical concentration when these compounds [e.g., substrates, cosubstrates, product(s)] may harm the enzyme.

## Methods

### Phylogenetic tree of ene reductases

The sequence space of ene reductases was analyzed with hhsuite[62] v3.1.0 using hhblits to search in uniclust30[63] with 12 starting sequences (YqiG, DrER, RmER, XenA, YqjM, ChrOYE1, NCR, XenB, NerA, OYE1 (Uniprot IDs can be found in Supplementary Table 1), EBP1 (Uniprot ID: P43084), Nox (Uniprot ID: A0A0N9HP11)). All 12 results were filtered by setting `-diff` flag to 50 to obtain the 50 most different sequences, which were then combined with the ene reductase sequences from literature[41,47]. The hits from the hhblits searches were filtered based on sequence similarity to the known ene reductases described in the two reviews[41,47] to reduce the number of sequence for the representation in phylogenetic tree. All sequences with a sequence similarity of less than 30% to any of the described ene reductases discarded and all others were aligned using Clustal Omega version 1.2.4-1 with default settings[64]. The resulting MSA was used to build a phylogenetic tree using FastTree version 2.1.10-1 with default settings[65]. After this analysis 13 ene reductases were arbitrarily picked across the phylogenetic tree to ensure sequence heterogeneity in the test case. The pairwise distance matrix of the selected 13 ene reductases was calculated using Clustal Omega with `--distmat-out` and `--full` flags.

### General experimental information

All chemicals were purchased from Sigma Aldrich and were used without further purification. For bacteria cultivation NaCl was purchased from Carl Roth GmbH + Co. KG and yeast extract and peptone were purchased from Thermo Fisher Scientific.

### Heterologous expression and purification of ene reductases and transaminases.

All ER candidates were expressed in *E. coli* BL21(DE3). An overnight culture (15 ml Lysogeny Broth [LB] media, 50 ng ml$^{-1}$ kanamycin [or 100 ng ml$^{-1}$ ampicillin for XenA, XenB, NerA]) was inoculated from a glycerol stock or LB plate and incubated at 37 °C and shaken at 120 rpm. For expression two times 600 ml of LB for each enzyme supplemented with appropriate antibiotics were inoculated from the overnight cultures for each candidate and shaken until the OD$_{600}$ reached $0.5 - 0.6$ (37 °C, 120 rpm). Then expression was induced

with isopropyl-$\beta$-D-thiogalactopyranoside (IPTG) (0.2 mM). Immediately after induction the temperature was reduced to 20 °C and the cultures were shaken overnight (120 rpm). Cells were harvested by centrifugation for 20 min at $4700 \times g$ (Hitachi himac-CR22N refrigerated centrifuge with a R10A3 rotor; 4 °C). Then the cell pellets were frozen at −20 °C.

For purification cell pellets (4–7 g) were resuspended in binding buffer (20 mM sodium phosphate, pH 7.4, 500 mM NaCl, 20 mM imidazole; 5 ml g$^{-1}$ wet mass of the pellet) supplemented with a spatula tip FMN and sonicated (program: 3.5 min, 2 s pulse, 4 s pause, amplitude: 30%). The soluble fraction was separated from the insoluble by centrifugation (Hitachi himac-CR22N refrigerated centrifuge with a R20A2 rotor; $43200 \times g$, 30 min, 4 °C). Prior to loading the soluble fraction onto an equilibrated HisTrap FF column (GE Healthcare, 5 ml) the soluble fraction was filtered through a 45 μm syringe filter. After loading the column was washed with binding buffer according to the manufacturer's manual. Then, the enzyme was eluted using elution buffer (20 mM sodium phosphate, pH 7.4, 500 mM NaCl, 500 mM imidazole). All purification steps were followed by SDS PAGE. The volume of the fractions containing the most enzyme (colored yellow) was reduced to 2.5 ml using a Vivaspin Ultrafiltration Unit (Satorius). Then, the buffer was changed to 50 mM sodium phosphate pH 7.4 using PD 10 Desalting Column (GE Healthcare). Approximately 3.5 ml of each enzyme with concentrations between 35 to 140 mg ml$^{-1}$ were obtained.

Transaminases ArS (internal plasmid number pEG29) and ArR (pEG234) were prepared according to published protocols[66,67] as follows: As an example for ArR, *E. coli* cells containing the overexpressed enzyme were disrupted using HEPES buffer (100 mM, pH 8) containing imidazole (20 mM) and PLP (1 mM). The same buffer was used for equilibrating the HisTag column, and also for the washing step after the crude extract was applied to the column. The ArR was eluted using the following buffer: 100 mM HEPES pH 8 containing 200 mM imidazole. After the protein containing fractions were pooled the puffer was changed (desalting PD10 column) to 10 mM HEPES pH 8. Aliquots of enzyme were stored at −20 °C until measurements.

The concentration of the purified enzymes was determined using a commercial BCA kit from Thermo Fisher according to the manual using the microplate procedure and BSA for the determination of the standard curve. Aliquots of enzyme were stored at −20 °C until melting curves and activities were measured.

### Measuring of melting temperature and unfolding of enzymes.

Thermal melting curves of enzymes (1 mg ml$^{-1}$) in aqueous buffer (50 mM sodium phosphate pH 7.4) supplemented with varying concentrations of organic co-solvents (dimethylsulfoxide, ethanol, methanol, 2-propanol, *n*-propanol) were recorded on a CFX96 Touch Real-Time PCR Detection System (Biorad) and a Prometheus NanoDSF system (Nanotemper).

The ThermoFMN assay was performed in a microplate (MLL9651, Biorad) sealed with Microseal'B' (MSB1001, Biorad) and was measured on CFX96 Touch Real-Time PCR Detection System (Biorad). After equilibration to 20 °C the temperature was increased with a gradient of 1 °C min$^{-1}$ to 90 °C. The change of the fluorescence of the cofactor FMN upon release was followed using the FAM channel [excitation (450–490 nm) and emission (510–530 nm) wavelength]. Melting temperatures were obtained by identifying the inflection point of the recorded curve using python 3. All ThermoFMN experiments were performed in triplicate. Estimated errors are derived from the standard error of the mean using Gaussian error propagation and given as error bars.

All nanoDSF experiments: Enzyme concentrations were between 0.5 to 2 mg ml$^{-1}$. Before measuring a sample, it was equilibrated at room temperature (20–22 °C) at the respective solvent condition for 16 h overnight. Capillaries were filled with ~10 μl sample and heated with a gradient of 2 °C min$^{-1}$ from 20 to 90 °C. The change of fluorescence of intrinsic aromatic residues upon unfolding was tracked at 330 nm and

350 nm. Analyzing the ratio of the spectroscopic signal (350 nm/330 nm) at a specific temperature in samples with an increasing amount of co-solvent gives evidence on the fraction of (un-)folded protein in the sample. The two-state model of unfolding[59,60] with the described empiric correction terms Eq. (1) was fitted to this data.

**Activity of enzymes.** The activity of the ene reductases (5–30 µg ml$^{-1}$) was analyzed with cyclohex-2-enone (10 mM) as model substrate in the presence of NAD(P)H (0.2 mM) in sodium phosphate buffer (50 mM, pH 7.4) and in a reaction volume of 200 µl in a microplate (655101, Greiner Bio-One) at the temperature indicated. For LacER, NADH was used as co-factor, for all others NADPH. The reaction was followed by measuring the absorbance of NAD(P)H at 340 nm on a plate reader (SpectraMax M2, Molecular Devices). During the optimization of the procedure the following order of adding the components was essential. Adding the enzyme first using a multi-stepper pipette (2–6 µl), followed by the addition of the needed volumes of distilled water (144–84 µl) and co-solvent (0–60 µl) using single-channel pipettes followed by the addition of substrate (20 µl of a 100 mM stock solution) and then sodium phosphate buffer from 10 x stock solution (16 µl) using multi-channel pipettes proved to give most reliable results. All stock solutions were prepared in 50 mM sodium phosphate buffer pH 7.4. The reaction was started by adding the co-factor with a multi-channel pipette (14 µl of a 2.7 mM stock solution) followed by an extra mixing step (150 µl pipetting up and down with a multi-channel pipette). The enzyme amount was determined such that the reaction could be followed for up to 240 s in the linear range. The linear part of the recorded curve was fitted and the slope and the extinction coefficient of NAD(P)H[68] were used to calculate the specific initial activity via Lambert-Beer's law. All initial activities are three-fold or five-fold determinations. Estimated errors are derived from the standard error of the mean using Gaussian error propagation and given as error bars.

**Activity at higher concentrations and set temperatures.** The experimental procedure was as described in the previous paragraph with a few modifications: The amount of enzyme was adjusted such that the linear range of the reaction could be followed for 480 s. The order of adding the components was the same: Enzyme was added with a multi-stepper pipette, next the 54–144 µl of distilled water were added with a multi-channel pipette. The desired amount of co-solvent (0–90 µl) 0–45% (v/v), the substrate (20 µl) and then sodium phosphate buffer from 10 x stock solution (16 µl) were added with a multi-stepper pipette. Before the reaction was started by the addition of the co-factor, the plate was sealed to avoid evaporation and pre-incubated until the samples reached the desired reaction temperature. The temperature was checked with a thermometer. Just before the start of the measurement, the co-factor (14 µl) was added with a multi-channel pipette. The reaction was mixed by thoroughly pipetting up and down, on top of that a 5 s shaking step was added to the protocol of the microplate reader before the start of the measurement and the reaction was followed for 480 s. A fresh seal was used to cover the samples during the measurement. The analysis of the data was performed as described above.

### Data processing, analysis, and model fitting
**General data handling and analysis.** All data processing, analysis and general data handling was done with python 3.7.10 using numpy 1.20.3 and pandas 1.2.4[69,70]. Regressions for calculation of experimental values were done with scikit-learn version 0.24.2[71]. The experimentally determined changes of melting temperature and initial activities in the presence of organic co-solvent was normalized using the value determined in aqueous environment without any co-solvent (native). This resulted in relative $T_m$ values between −1 and 0. Relative activities are given in percent. The relative activities reach values above 100% because the addition of co-solvents led to an increased initial activity in some cases. All plotting was done using matplotlib version 3.4.2[72].

**Model development and fitting.** Both, the fitting of the two-state unfolding model and the suggested activity solvent model were performed using the minimize function of the scipy python package version 1.5.3[73]. The unfolding model Eq. (1) was implemented as previously described using the sum of least squares as cost function during the minimization. The parameters $\alpha_F$, $\beta_F$, $\alpha_U$, $\beta_U$, $m_{folding}$ and $c_{U_{50}}$ were first scaled to the same order of magnitude and then optimized. The lower bound for all six fit parameters was set to 0 and an upper bound of 100 was only set for $c_{U_{50}}$. The universal gas constant $R$ was set to 8.314 J mol$^{-1}$ K$^{-1}$ and $T$ was set to the respective temperature of the studied system in Kelvin.

The fitting of the activity solvent model in Eq. (3) was implemented likewise. In contrast however an L2 regularization was used as cost function. The regularization is a sum of weighted squares of the parameter values. The parameters $\xi$, $\nu$, $\sigma$, $c_{A_{max}}$, $m_{folding}$ and $c_{A_{50}}$ were optimized. In this case, scaling was necessary only for $m_{folding}$ to avoid biases during the minimization. The lower bounds for $\xi$, $\nu$, $c_{A_{max}}$ and $c_{A_{50}}$ were set to 0. For $\sigma$ a lower bound of 10 was applied. For $c_{A_{max}}$ and $c_{A_{50}}$ upper bounds of 100 were used. In all other cases no bounds were used. During the regularization $\lambda$ was set to 0.01 for $c_{A_{max}}$ and to 0.001 for $\xi$. In all other cases no extra L2 regularization was added. The universal gas constant $R$ was set to 8.314 J mol$^{-1}$ K$^{-1}$ and $T$ was set to the respective temperature of the studied system in Kelvin.

**Correlation analysis.** Correlation analysis was done using stats.pearson from scipy python package version 1.5.3[73]. The relative specific activity with the change of melting temperature was analyzed as discussed in the main text. For the correlation analysis of $c_{A_{50}}$ and $c_{U_{50}}^T$, all experiments where both the two-state unfolding model and the activity model could successfully be fitted were included. Additionally, to the limitations imposed by lacking activity data which was discussed in the main text, the values for NCR in the presence of ethanol had to be excluded because the two-state model of unfolding could not fit the recorded unfolding.

### Reporting summary
Further information on research design is available in the Nature Portfolio Reporting Summary linked to this article.

## Data availability
All methods and the data that support the findings of this study are available in the Supplementary Information of this article. All data generated in this study are provided in the Supplementary Information/Source Data file. Source data are provided with this paper.

## Code availability
The custom functions written for the analysis of the data contained in the manuscript are deposited and publicly available at the repository: https://github.com/basf/EnzSolvStab[74].

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

## Acknowledgements
The COMET center: acib: Next Generation Bioproduction is funded by BMK, BMDW, SFG, Standortagentur Tirol, Government of Lower Austria and Vienna Business Agency in the frame work of COMET—Competence Centers for Excellent Technologies. The COMET-Funding Program is managed by the Austrian Research Promotion Agency FFG. The University of Graz and the Field of Excellence BioHealth are acknowledged for financial support. I.S. Sorgenfrei is acknowledged for advice on minimization and regularization.

## Author contributions
Design of the work: F.A.S., D.S., S.S., and W.K. Data acquisition: F.A.S., J.J.S., N.A.M., F.W., and M.Z. Data analysis: F.A.S. and S.S. Data interpretation: F.A.S., T.L.E., J.J.S., S.S., and W.K. Manuscript writing: F.A.S., S.S., and W.K.

## Competing interests
F.A.S., F.W., M.Z., S.S. and W.K. are authors of a patent application: European Patent application 23165017.7 filed on 29 Mar 2023. The remaining authors declare no competing interests.
