## [Peer Review File · Nature Communications]

Solvent concentration at 50% protein unfolding may reform enzyme stability ranking and process window identificationREVIEWER COMMENTS

Reviewer #1 (Remarks to the Author):

The authors sought to find a criterion that indicates a transition, most easily thought to be one from native, folded protein to inactive, unfolded one, in aqueous-miscible organic solvent mixtures. They argue that the melting temperature T_m , commonly used for protein transitions in aqueous medium, is not a good indicator for transitions in aqueous-organic mixtures. While this rationale is correct and the objective is very worthwhile, the authors did not deliver on that promise.

1. C,T,U50 is not a simple criterion to be utilized widely in the community. While the authors correctly place the transition in aqueous-organic mixtures at 50% unfolded (and 50% folded) protein (analogous to T_m), the inclusion of temperature as a parameter is unnecessarily cumbersome. While T_m also depends on the composition of the medium, such as with kosmotropes and chaotropes, it is measured and reported simply as T_m , and often measured at comparably standard conditions, such as in PBS buffer. Thus, C,U50 would completely suffice, provided it were to be measured at a standard temperature, say 25 or 30°C. Measurement at a standard temperature would actually result in larger amounts of organic solvent to be added to reach the spot, where $[N] = [U]$. Thus, the inclusion of temperature as a parameter in fact obscures the impact of different organic solvents.

2. As Figure 7 demonstrates, C,T,U50 is not a very reliable criterion for prediction of enzyme stability in organic solvents. The correlations are rather weak, and this concerns the otherwise well-behaved solvents methanol and ethanol.

3. The choice of solvents was not very adventurous: while DMSO is a good choice, the pick of 4 structurally similar alcohols limits the value of the study. Inclusion of solvents such as DMF and THF, preferably as 6th and 7th solvent rather than 4th and 5th, would have been more interesting.

4. The authors ignore or are not cognizant of all of the relevant literature. A criterion, the denaturing capacity DC, has been described for enzyme stability in aqueous-organic

mixtures (Khmelnitsky, Y. L.; Mozhaev, V. V.; Belova, A. B.; Sergeeva, M. V.; Martinek, K. Eur. J. Biochem. 1991, 198, 31). The authors should have considered this work and compared it with their own.

5. Most importantly: the authors cherrypicked their data on four EREDs (ChrOYE1, OYE1, PpXenB, and TsOYE) of the 13 in the main text to make claims that are less firm or are strongly diminished when considering even all 13 ERED enzymes (not to say anything of ore EREDs or other enzymes) with data in the SI. As an example, SI Figure 6 shows F350/F330 data on the y-axis plotted against solvent content, here ethanol. Only 7 of the 13 EREDs show any usable transition! The applicability of the C,T,U50 approach to other systems, even other EREDs, seems tenuous at best and will not be approached with confidence.

For all these, and more, points, this manuscript should not be accepted.

Reviewer #2 (Remarks to the Author):

Dear Authors

The manuscript Sorgenfrei et al. entitled „Solvent concentration at point of 50% protein unfolding **cU50 T** may reform enzyme solvent stability ranking and identification of process window“ describes a novel approach to identify conditions for applying enzymes in the presence of water miscible co-solvents. Often the melting temperature is or was used to get a hint towards the process stability of enzymes also in the presence of solvents. However, as the authors state this is not always leading to a success, and one needs another more accurate routine to predict this process window. This important issue was now tackled by the authors and as role model they employed ene reductases. For this class of enzymes, a huge set of data are readily available from literature including thermo- as well as solvent stability, making it an ideal candidate for such an approach to establish a new routine to predict process applicability of enzymes in the presence of miscible co-solvents.

Furthermore, the authors laboratories have a long-lasting experience with this enzyme family. All this expertise was used to generate a holistic new set of data to establish a new key parameter to evaluate enzyme stability towards solvents connected to enzyme performance.

This new concept will be of use for the community, especially in biocatalysis, but it may be of interest to a broader audience as well. Protein stability and folding is a very diverse field, and such supportive methods will help to uncover novel features among those molecules and their function. Thus, I can state this is a novel and original contribution to the field of high impact.

A hypothesis was formulated, and proper methods have been employed to answer the raised questions. The methods are well documented and sound reproducible. Needed standards to describe enzymes have been applied and especially for flavin dependent ene reductases. Those can be transferred to other enzymes as well.

In general, the manuscript is well prepared, scientific sound and all needed information are provided to follow the study. A new concept is presented and supported by a large set of data and well discussed. Proper conclusions were drawn from a sophisticated data analysis. The reference list is appropriate and most relevant articles are cited; no excess self-citation determined. General formatting is of good quality and presentation of tables and figures very supportive. Nevertheless, some minor points need to be improved – nothing critical from my point of view.

In the following I will provide my comments and hope they are supportive and constructive:

- This part of the sentence from line 24 is not sound: “showing depending also on the solvent itself” and might be rephrased.
- Line 78/79 while I agree that FMN is a true cofactor; NAD(P)H is rather a co-substrate for EREDs as it is consumed during catalysis and cannot be recycled by the enzyme in the same reaction course.
- The general observation towards T_m that DMSO is less effective while the alcohols are more effective as the chain length increases is quite interesting; was to some extent observed for oxidases as well (of course not that generalized as shown herein); it might have to do with the change in pKa and acidity of the HO-group. Can you comment on this?
- Line 144 “spectroscopically” rather “spectrophotometrically”
- Line 211 Add a space “(a) The ...”

- Line 263 as well as Line 275 and Line 285: correct "Equation ((3)." to "Equation (3)."
- Line 398, correct "sequence"
- Line 425, correct "temperature"
- Ene reductase quantification; did you determine the FMN loading? As not all ene reductases are fully loaded, and what was used to set protein concentration (active sites; FMN-loaded or total protein?); specify.

Supplementary Material:

- The given activities are mostly presented as relative values; please indicate the 100% as U mg⁻¹ or observed rate as s⁻¹.
- SFIG3.: insert a space "pH 7.4using" to correct "pH 7.4 using"
- SFIG4.: "cofactor" delete "-"

General point: protein folding and unfolding is studied from many perspectives. Often unfolding curves are determined by titrating guanidine HCl or urea. I just wonder, if one could expect similar effects here as you observed by comparing unfolding and activity? From a protein point of view, you change in both cases the environment by a chemical agent leading to partial and later to total unfolding. I just wonder if your concept can be extended to other agents as well.

Reviewer #3 (Remarks to the Author):

This paper investigates parameters that influence enzyme activity in organic water miscible solvents. There is significant amount of literature suggesting that thermal stability can be taken as a surrogate for tolerance of organic solvent, but the authors show that the correlation is not always strong. They suggest that as an alternative the stability of the enzyme with respect to increasing amount of organic solvent should be used. They also show that solvent stability differs between solvents. Overall, these results are of interest in biotechnology, where organic solvents are often needed to solubilise hydrophobic substrates and allow for high substrate concentrations.

The study is very thorough with many variations of parameters considered and the

conclusions are well justified.

My concern is with respect to the enzyme class selected - only EREDs were investigated, but the conclusions were drawn much more broadly. I would like to see data for other systems - with different protein folds and different co-factors. I assume all EREDs have the same fold? Is it possible that the data are due to NADPH binding pockets being influenced by solvent? The authors should comment on that.

It would be interesting to compare closely related variants with different solvent tolerance and /or temperature stability.

Point by point answer

We thank all reviewers for the valuable comments to help us to improve our manuscript.

REVIEWER COMMENTS

Reviewer #1 (Remarks to the Author):

The authors sought to find a criterion that indicates a transition, most easily thought to be one from native, folded protein to inactive, unfolded one, in aqueous-miscible organic solvent mixtures. They argue that the melting temperature T_m , commonly used for protein transitions in aqueous medium, is not a good indicator for transitions in aqueous-organic mixtures. While this rationale is correct and the objective is very worthwhile, the authors did not deliver on that promise.

1. $C_{T,U50}$ is not a simple criterion to be utilized widely in the community. While the authors correctly place the transition in aqueous-organic mixtures at 50% unfolded (and 50% folded) protein (analogous to T_m), the inclusion of temperature as a parameter is unnecessarily cumbersome. While T_m also depends on the composition of the medium, such as with kosmotropes and chaotropes, it is measured and reported simply as T_m , and often measured at comparably standard conditions, such as in PBS buffer. Thus, C_{U50} would completely suffice, provided it were to be measured at a standard temperature, say 25 or 30°C. Measurement at a standard temperature would actually result in larger amounts of organic solvent to be added to reach the spot, where $[N] = [U]$. Thus, the inclusion of temperature as a parameter in fact obscures the impact of different organic solvents.

Response: We thank the reviewer for the feedback. Maybe it was not clear from the manuscript how the $c_{U_{50}}^T$ was determined. The description was now revised making clear that temperature is intrinsically involved. Nevertheless, the data also show, that the temperature has a significant impact at which solvent concentration each enzyme is unfolding. Therefore, the temperature is of importance. The text in the manuscript was adapted as follows to clarify the mode of measurement:

To obtain $c_{U_{50}}^T$ values for enzyme/co-solvent combinations, samples containing the same enzyme at varied concentrations of solvent were followed via nano differential scanning fluorimetry (nanoDSF) experiments, thus measuring the unfolding at increasing temperature. NanoDSF is used to track the fluorescent signal arising from the internal tryptophan/tyrosine residues and has previously been used to identify melting points of enzyme libraries.⁵⁸ From multiple nanoDSF experiments the $c_{U_{50}}^T$ can be extracted by plotting the unfolding at a specific temperature versus solvent concentration (**Fig. 4a** and **Fig. 5a**). Consequently,...

Indeed, the analysis of this value at a (to be) defined standard temperature yields already information on the solvent tolerance, however, as other parameters (like temperature, but also any other salt/ingredients) influence the conditions, the measurement conditions will always depend on a specific case/conditions. Comparison can of course only be done using the same conditions. We agree that in case no nanoDSF machine is available and more tedious methods for the collection of the unfolding data like DSF has to be performed, just analyzing specific condition can be advisable. Anyway, it may be highly beneficial to have a broader set of stability data also at varied temperature to find optimal operational windows as exemplified in the paper. Thus, looking at more temperature and solvent concentrations together we provide a strategy suitable for the optimization of reaction conditions close to practical requirements.

Reviewer #1

2. As Figure 7 demonstrates, C,T,U50 is not a very reliable criterion for prediction of enzyme stability in organic solvents. The correlations are rather weak, and this concerns the otherwise well-behaved solvents methanol and ethanol.

Response: We think that the conclusion of the reviewer goes in the wrong direction and contradicts his first own statements (“While the authors correctly place the transition in aqueous-organic mixtures at 50% unfolded (and 50% folded) protein..”) that the unfolding is a suitable parameter for stability. What is probably easier spotted from Figures 4/5 is that the inflection point for activity correlates with the inflection point of unfolding; this is more difficult recognized in the further processed data for the activity. We set up a model for the activity considering the boosting effect via a Gaussian term, which is only an approximation. Additionally, the activity data in Figure 4/5 shows significant error bars which leads to moderate fitting. We tried to use this activity model to compare the inflection points of activity and unfolding data. We agree that the correlation could be better, but this indicates, that either the model needs further refinement or the error of the activity measurements are significant. This does not influence the overall story. To address this point by the reviewer the following sentence was added:

Nevertheless, the correlation values may indicate how difficult it is to measure activities with low errors and that maybe a further improved activity model may be required.

Reviewer #1

3. The choice of solvents was not very adventurous: while DMSO is a good choice, the pick of 4 structurally similar alcohols limits the value of the study. Inclusion of solvents such as DMF and THF, preferably as 6th and 7th solvent rather than 4th and 5th, would have been more interesting.

Response: We are grateful for the input and have now included additional solvents to this study. Furthermore, we have also added data for two transaminases.

Finally, to show the applicability also for other solvents than the ones just mentioned (MeOH/EtOH), $c_{U_{50}}^T$ values were also measured for selected EREDs in DMSO, DMF and *n*-propanol (**Table 3, Supplementary Table 4, Supplementary Fig. 12**). THF was also tested, but it was noticed, that it was not tolerated by the enzymes investigated. From the data obtained (e.g. entries 1-4) it can nicely be seen that the $c_{U_{50}}^T$ values change with the temperature, thus, the temperature has a clear impact. In general, the higher the temperature, the lower the $c_{U_{50}}^T$ value. Additional experiments also indicated the applicability to other enzymes like transaminases (entries 17-24). The two transaminases investigated, one (*S*)-selective one originating from *Arthrobacter citreus* (ArS) and the other (*R*)-selective one from an *Arthrobacter* sp. (ArR) possess different structural folds. Also for these enzymes $c_{U_{50}}^T$ values were successfully determined.

Table 3 Examples for $c_{U_{50}}^T$ values for various EREDs in further solvents as well as $c_{U_{50}}^T$ values for two transaminases (ArR, ArS). The complete table can be found in the SI (Supplementary Table 4).

Entry	Enzyme	solvent	T [°C]	$c_{U_{50}}^T$ ^a [% (v/v)]	Entry	Enzyme	solvent	T [°C]	$c_{U_{50}}^T$ ^a [% (v/v)]
1	ChrOYE1	DMF	25	23	13	TsOYE	DMSO	70	41
2	ChrOYE1	DMF	30	15	14	TsOYE	nprop	45	18
3	ChrOYE1	DMF	35	13	15	YqiG	DMF	45	35
4	ChrOYE1	DMF	40	11	16	YqiG	nprop	25	18
5	ChrOYE1	DMSO	45	41	17	ArR	nprop	30	18
6	ChrOYE1	nprop	30	10	18	ArR	nprop	35	18
7	ChrOYE1	nprop	35	8	19	ArR	nprop	70	11
8	ChrOYE1	nprop	40	7	20	ArS	DMF	60	31
9	DrER	DMF	30	23	21	ArS	DMSO	60	45
10	DrER	DMSO	30	33	22	ArS	EtOH	50	25
11	DrER	nprop	30	14	23	ArS	MeOH	45	47
12	TsOYE	DMF	60	37	24	ArS	nprop	45	23

^a The error of the $c_{U_{50}}^T$ was estimated to be about 2%.

In the Experimental section:

Transaminases ArS (internal plasmid number pEG29) and ArR (pEG234) were prepared as previously described.^{66,67} As an example for ArR, cells were disrupted using HEPES buffer (100 mM, pH 8) containing imidazol (20 mM) and PLP (1 mM). The same buffer was used for equilibrating the HisTag column, and also for the washing step after the crude extract was applied to the column. The ArR was eluted using the following buffer: 100 mM HEPES pH 8 containing 200 mM imidazol. After the protein containing fractions were pooled the puffer was changed (desalting PD10 column) to 10 mM HEPES pH 8. Aliquots of enzyme were stored at -20 °C until measurements.

66. Pressnitz, D. *et al.* Asymmetric amination of tetralone and chromanone derivatives employing

omega-transaminases. *ACS Catal.* **3**, 555–559 (2013).

67. Alvarenga, N. *et al.* Asymmetric Synthesis of Dihydropinidine Enabled by Concurrent Multienzyme Catalysis and a Biocatalytic Alternative to Krapcho Dealkoxycarbonylation. *ACS Catal.* **10**, 1607–1620 (2020).

Reviewer #1

4. The authors ignore or are not cognizant of all of the relevant literature. A criterion, the denaturing capacity DC, has been described for enzyme stability in aqueous-organic mixtures (Khmelnitsky, Y. L.; Mozhaev, V. V.; Belova, A. B.; Sergeeva, M. V.; Martinek, K. *Eur. J. Biochem.* 1991, 198, 31). The authors should have considered this work and compared it with their own.

Response: We thank the reviewer for pointing out this reference and have included it in the introduction and have compared the opportunities the here presented CTU50 parameter gives with the options of the in the reference described denaturation capacity.

An alternative parameter for the selection of an organic solvent for a biotransformation is the denaturation capacity of a solvent as guidance across different enzymes³³.

Due to these differences, $c_{U_{50}}^T$ is worth to be considered in the ranking of enzymes. Compared to the denaturation capacity parameter³³, our approach also allows for a ranking of different enzymes in dependence of the solvent and not a ranking of co-solvents only. Especially when looking for the best enzyme of a library or analyzing variants in a protein engineering campaign,...

33. Khmelnitsky, Y. L., Mozhaev, V. V., Belova, A. B., Sergeeva, M. V. & Martinek, K. Denaturation capacity: a new quantitative criterion for selection of organic solvents as reaction media in biocatalysis. *Eur. J. Biochem.* **198**, 31–41 (1991).

Reviewer #1

5. Most importantly: the authors cherry-picked their data on four EREDs (ChrOYE1, OYE1, PpXenB, and TsOYE) of the 13 in the main text to make claims that are less firm or are strongly diminished when considering even all 13 ERED enzymes (not to say anything of ore EREDs or other enzymes) with data in the SI. As an example, SI Figure 6 shows F350/F330 data on the y-axis plotted against solvent content, here ethanol. Only 7 of the 13 EREDs show any usable transition! The applicability of the C,T,U50 approach to other systems, even other EREDs, seems tenuous at best and will not be approached with confidence.

Response: Not surprisingly, we strongly disagree with this statement. We displayed only four representative examples in the main paper to improve the readability and not for cherry-picking. All data is in the SI to which we refer to several times and we also discuss data of enzymes in detail that are only shown in the SI e.g. Tm of NerA in lines 103 + 118 (original version). In line 207-211 (original version) we also state that some enzymes gave unfolding data that was not possible to analyse in the described way. Line 252 states where to find more data and unfolding and activity fits and which SI Figures need to be compared to see the relation of the unfolding and the loss of the activity around the same co-solvent concentration. On top of that, for all values given, they were in all cases calculated from all recorded data, that could be included. In the main text it is always explained in detail, which data series were combined and which criteria were used to make this decisions. We actually mentioned limitations and reasons why seven could be interpreted (For some enzymes this method did not give

an analysable unfolding curve (e.g. LacER, see **Supplementary Fig. 6** and **Supplementary Fig. 7**), which was attributed for example to tryptophan residues with little change in their local environment during the unfolding process.). This is a limit of the analytical method, not of the C,T,U50.

Reviewer #1

For all these, and more, points, this manuscript should not be accepted.
Response: We thank the reviewer for the comments especially to include also the DC, which we have now done; we clarified that the measurement for $c_{U_{50}}^T$ goes in hand with measuring the melting temperature in a very simple set up using nanoDSF, and also clarified that we did not perform cherry picking but showed representative data in the paper to make it easier to read and underlined that all data is in the SI.

Reviewer #2 (Remarks to the Author):

Dear Authors

The manuscript Sorgenfrei et al. entitled „Solvent concentration at point of 50% protein unfolding **cU50 T** may reform enzyme solvent stability ranking and identification of process window“ describes a novel approach to identify conditions for applying enzymes in the presence of water miscible co-solvents. Often the melting temperature is or was used to get a hint towards the process stability of enzymes also in the presence of solvents. However, as the authors state this is not always leading to a success, and one needs another more accurate routine to predict this process window. This important issue was now tackled by the authors and as role model they employed ene reductases. For this class of enzymes, a huge set of data are readily available from literature including thermo- as well as solvent stability, making it an ideal candidate for such an approach to establish a new routine to predict process applicability of enzymes in the presence of miscible co-solvents. Furthermore, the authors laboratories have a long-lasting experience with this enzyme family. All this expertise was used to generate a holistic new set of data to establish a new key parameter to evaluate enzyme stability towards solvents connected to enzyme performance.

This new concept will be of use for the community, especially in biocatalysis, but it may be of interest to a broader audience as well. Protein stability and folding is a very diverse field, and such supportive methods will help to uncover novel features among those molecules and their function. Thus, I can state this is a novel and original contribution to the field of high impact.

A hypothesis was formulated, and proper methods have been employed to answer the raised questions. The methods are well documented and sound reproducible. Needed standards to describe enzymes have been applied and especially for flavin dependent ene reductases. Those can be transferred to other enzymes as well.

In general, the manuscript is well prepared, scientific sound and all needed information are provided to follow the study. A new concept is presented and supported by a large set of data and well discussed. Proper conclusions were drawn from a sophisticated data analysis. The reference list is appropriate and most relevant articles are cited; no excess self-citation determined. General formatting is of good quality and presentation of tables and figures very supportive. Nevertheless, some minor points need to be improved – nothing critical from my point of view.

Response: We thank the reviewer for acknowledging the impact of the study and also for the supportive statements.

Reviewer #2: In the following I will provide my comments and hope they are supportive and constructive:

- This part of the sentence from line 24 is not sound: “showing depending also on the solvent itself” and might be rephrased.

Response: The sentence was rephrased:

Comparing possible rankings of enzymes according to melting temperature and $c_{U_{50}}^T$ revealed a clearly diverging outcome **also depending** on the **specific solvent used**.

Reviewer #2: - Line 78/79 while I agree that FMN is a true cofactor; NAD(P)H is rather a co-substrate for EREDs as it is consumed during catalysis and cannot be recycled by the enzyme in the same reaction course.

Response: We have rephrased the sentence and refer to NAD(P)H as cosubstrate now:

‘For this study, ene-reductases (EREDs) using the cofactors FMN and NAD(P)H **as cosubstrate** were selected as model catalysts possessing high relevance for biocatalysis³⁴⁻⁴⁰.’

Reviewer #2: - The general observation towards T_m that DMSO is less effective while the alcohols are more effective as the chain length increases is quite interesting; was to some extent observed for oxidases as well (of course not that generalized as shown herein); it might have to do with the change in pKa and acidity of the HO-group. Can you comment on this?

Response: We like very much the idea to link the trend of T_m to the pKa, although when we compared the order of effect with the pKa values, there is not a perfect fit: Methanol (pKa 15.2) < ethanol (pKa 16) < 2-propanol (pKa 17.1) < n-propanol (pKa 16.85). Probably the pKa is one parameter to consider maybe together also with volume... There is more research needed. As we do not spot a clear relation, we did not put any speculations to the manuscript.

Reviewer #2

- Line 144 “spectroscopically” rather “spectrophotometrically”
- Line 211 Add a space “(a) The ...”
- Line 263 as well as Line 275 and Line 285: correct “Equation ((3).” to “Equation (3).”
- Line 398, correct “sequence”
- Line 425, correct “temperature”

Response: For all cases the text was modified as requested and are now highlighted with yellow background in the main manuscript.

Reviewer #2

- Ene reductase quantification; did you determine the FMN loading? As not all ene reductases are fully loaded, and what was used to set protein concentration (active sites; FMN-loaded or total protein?); specify.

Response: Additional FMN was added to the preparation during cell disruption. The concentration of all EREDs was determined after purification using BCA assay. We have added the following to the Methods section:

The concentration of the purified enzymes was determined using a commercial BCA kit from Thermo Fisher according to the manual using the microplate procedure and BSA for the determination of the standard curve.

Reviewer #2

Supplementary Material:

- The given activities are mostly presented as relative values; please indicate the 100% as U mg⁻¹ or observed rate as s⁻¹.

Response: All raw data including absolute values for the specific activities are given in **Supplementary Table 5 and 6**. We have added a remark to all Figures with relative activities.

Note: All raw data that was used to obtain the derived values ΔT_m and relative specific activity can be found in **Supplementary Table 5 and 6**.

Reviewer #2

- SFIG3.: insert a space “pH 7.4using” to correct “pH 7.4 using”
- SFIG4.: “cofactor” delete “-“

Response: Both Supplementary Figure captions were corrected. The adaptations are highlighted now with yellow background.

Reviewer #2

General point: protein folding and unfolding is studied from many perspectives. Often unfolding curves are determined by titrating guanidine HCl or urea. I just wonder, if one could expect similar effects here as you observed by comparing unfolding and activity? From a protein point of view, you change in both cases the environment by a chemical agent leading to partial and later to total unfolding. I just wonder if your concept can be extended to other agents as well.

Response: Thank you for this very interesting and constructive remark. Indeed, the theoretical process of unfolding induced by cosolvents as described in this study and the unfolding induced by other agents like guanidine HCl or urea or any other inorganic/organic compound are expected to be similar. The detailed steps of unfolding induced by either the cosolvents or the alternative agents might differ between the other agents but also between the different cosolvents. This behaviour however has not been analysed with in this study. Anyway, it is rather straightforward to assume that this will work. Consequently, we added the following text to the outlook:

It is fair to assume that the applicability of the method can also be extended to any other organic compound and or inorganic component present, thus, allowing to determine the critical concentration when these compounds [e.g. substrates, cosubstrates, product(s)] may harm the enzyme.

Reviewer #3 (Remarks to the Author):

This paper investigates parameters that influence enzyme activity in organic water miscible solvents. There is significant amount of literature suggesting that thermal stability can be taken as a surrogate for tolerance of organic solvent, but the authors show that the correlation is not always strong. They suggest that as an alternative the stability of the enzyme with respect to increasing amount of organic solvent should be used. They also show that solvent stability differs between solvents. Overall, these results are of interest in biotechnology, where organic solvents are often needed to solubilise hydrophobic substrates and allow for high substrate concentrations.

The study is very thorough with many variations of parameters considered and the conclusions are well justified.

My concern is with respect to the enzyme class selected - only EREDs were investigated, but the conclusions were drawn much more broadly. I would like to see data for other systems - with different protein folds and different co-factors. I assume all EREDs have the same fold? Is it possible that the data are due to NADPH binding pockets being influenced by solvent? The authors should comment on that.

Response: We thank the Reviewer for this comment and have done in the meantime a significant amount of new measurements (now Table 3 and Supplementary Table 4) and added this data to the manuscript. As described for reviewer 1 at point 3, we included on the one had more/other solvents and also extended the examples to other type of enzymes, namely two transaminases, each with a different fold. As expected (at least for us), the concept was also applicable for the other type of enzymes. The stability depends on the structural elements and also whether the cofactor is bound or not bound but we did not investigate this point here. We have now included the text for the additional enzymes (two transaminases), each with a different fold.

Finally, to show the applicability also for other solvents than the ones just mentioned (MeOH/EtOH), $c_{U_{50}}^T$ values were also measured for selected EREDs in DMSO, DMF and *n*-propanol (**Table 3, Supplementary Table 4, Supplementary Fig. 12**). THF was also tested, but it was noticed, that it was not tolerated by the enzymes investigated. From the data (entries 1-4) obtained it can nicely be seen that the $c_{U_{50}}^T$ values change with the temperature, thus, the temperature has a clear impact. In general, the higher the temperature, the lower the $c_{U_{50}}^T$ value. Additional experiments also indicated the applicability to other enzymes like transaminases (entries 17-24). The two transaminases investigated, one (*S*)-selective one originating from *Arthrobacter citreus* (ArS) and the other (*R*)-selective one from an *Arthrobacter* sp. (ArR) possess different structural folds. Also for these enzymes $c_{U_{50}}^T$ values were successfully determined.

As the principles of this strategy are applicable for all enzymes having tryptophane/tyrosine in their sequence, this approach is not limited to EREDs and can be used for other types of enzymes, **as we have shown here also for transaminases**, and could have a major impact on the description of enzyme stability, selection of suitable enzymes from (commercial) libraries and in enzyme engineering campaigns, as well as on identifying operational windows for reactions.

Reviewer #3

It would be interesting to compare closely related variants with different solvent tolerance and /or temperature stability.

Response: Yes, we agree that it would be very interesting to analyse the solvent tolerance and temperature stability with respect to relation of the variants and we expect that this is a future application for the $c_{U_{50}}^T$. Consequently, we think that for a detailed analysis another study with variants is needed. The candidates of this study were picked widely to cover most of the sequence space of the known EREDs and was expanded by two transaminases to show the effect on another class of enzymes as well. The intention was to implement a method and indeed we agree that there are many more options for applications.

Additional changes not related to comments of reviewers

Due to maternity (Frieda Sorgenfrei) and paternity (Jeremy J. Sloan) leaves, we had a delay in answering and also “new” people did the nanoDSF measurements at BASF and were responsible for that. Consequently, the list of authors was extended by **Niklas A. Mehner², Thomas L. Ellinghaus²**.

Furthermore, we spotted several typos, which were now corrected and are marked in the main paper with a yellow background.

Finally we added:

Data availability

All methods and the data that support the findings of this study are available in the supplementary information of this article **and in the provided excel files containing the raw data.**

REVIEWERS' COMMENTS

Reviewer #2 (Remarks to the Author):

Dear Authors

as pointed out in my previous review, the presented results and concept are supportive for the biocatalysis community as well as for related field.

I can state, that you addressed all my requests and in addition the concerns raised by the other reviewers were also considered and overall this lead to an improved manuscript.

The addition of two other enzymes and solvents made it more strong and it seems truly applicable for various systems.

I have no further comments and congratulate to this large data collection which allowed to introduce a novel concept for biocatalysis.

The comments of reviewer 1 were very constructive and the authors addressed those sufficiently from my point of view.

In detail:

- some phrases in the original manuscript were not sufficient, the re-writing did improve and clarify those aspects. This was an important change in the manuscript especially for the methods and how they are applied.

- the concept was questioned for solvents and enzymes; and here the authors added both, hence new data were incorporated for two solvents and for another enzyme class. That was and is for me the most important point. That made the concept clear and wider applicable.

Hence, this drastically improved the manuscript.

- there was one comment from reviewer 1 on "cherry picking" the best candidates. I can see both arguments from reviewer 1 as well as from the authors. But, I agree with the authors: all data are cited in the text and it is for the general reader easier to see the applicability of the novel concept with drastic/obvious examples. Since the concept is critically discussed and it is clearly stated that it provides a chance to be developed, I see this rather as a chance and not as a drawback. From my point of view, this is well phrased and clear to the audience.

Overall; I can state that the authors made a huge effort to answer the comments from reviewer 1 and from my point of view this is very convincing. Thus, the comments from reviewer 1 are sufficiently addressed.

Reviewer #2 (Remarks on code availability):

I did check for the website and it seems to be functional; but the code itself I did not check.

Reviewer #3 (Remarks to the Author):

The paper develops novel parameters to assess operability of enzymes in organic solvents - as biocatalysis becomes more and more integrated into chemical processes, such parameters will be very valuable to assess biocatalysts from natural sources or from mutagenesis campaigns. As chemical substrates are often not very water soluble, organic co-solvents will be required.

The authors have done a very extensive study over a large number of proteins. In response to referees' comments they have expanded their work from one class of enzymes (EREDs) to others (eg transaminases) which validates the approach. The data are excellent both in terms of quantity and quality. All referees' comments have been carefully and satisfactorily addressed. This manuscript will make a very important contribution to this rapidly emerging field, where very large numbers of biocatalyst can be produced and need to be assessed using reliable biophysical methods.